# Nutritional Quality of Fast Food Kids Meals and Their Contribution to the Diets of School-Aged Children

**DOI:** 10.3390/nu12030612

**Published:** 2020-02-26

**Authors:** Ana Gabriela Palos Lucio, Diana Nicte-Há Sansores Martínez, Corina Olvera Miranda, Laura Quezada Méndez, Lizbeth Tolentino-Mayo

**Affiliations:** 1Facultad de Enfermería y Nutrición. Universidad Autónoma de San Luis Potosí, Niño Artillero 130, Zona Universitaria, 78240 San Luis Potosí, Mexico; 2Nutrition and Health Research Center, Instituto Nacional de Salud Pública, Av. Universidad 655, Santa María Ahuacatitlán, 62100 Cuernavaca, Morelos, Mexico; 3Facultad de Ciencias Químicas. Universidad Autónoma de San Luis Potosí, Av. Dr. Manuel Nava 608, Bellas Lomas, 78210 San Luis Potosí, Mexico; 4El Colegio de Chihuahua, Partido Díaz 4723, Progresista, 32310 Cd Juárez Chih., Mexico

**Keywords:** nutrition, fast food restaurants, caloric contribution, macronutrients, nutritive value, school age, children, teenagers, overweight, obesity

## Abstract

International data indicate that children and adolescents obtain around one third of their daily food intake from products consumed outside their home. Food products from restaurants are usually higher in calories and lower in nutritional value than those prepared home. We evaluated the nutritional quality in kids’ meals from three fast food chains and two movie theaters and compared them with nutritional recommendations for Mexican school-age population. Results showed that the menu options marketed for school-age children have higher caloric contributions than those recommended, in addition to a deficient nutritional quality. The contribution of caloric and of almost all macronutrients for all mealtimes is not only high but even above 100% or 200% of the mean recommended daily intake (reaching to more than 400% of the recommendations of carbohydrates and lipids of preschool age group). In particular, the snack main dish (popcorn), provides over 100% of the mean energy intake recommendations for the three school age groups and for preschool age group, this contribution could reach to 270%. Therefore, regulations regarding nutritional recommendations should exist for these types of commercialized food products for school-age children, along with mandatory and clear labeling that allows consumers to be able to make better choices for their kids.

## 1. Introduction

In 2016, Mexico declared an epidemiologic emergency due to high prevalence of obesity and diabetes mellitus [1]. While 13.6% of kids under 5 years old showed chronic malnutrition, 33.2% of children (5–11 years) and 36.3% of adolescents (12–19 years) were overweight or obese [2]. In addition to impacts on health, the economic burden represents 2.3% of Mexico’s gross domestic product (GDP) [3]. In Mexico, according to the Global Burden of Disease Study (GBD) (2010), 6%, 28%, and 62% of deaths related to cancer, diabetes, and cardiovascular diseases, respectively, are due to dietetic risk factors such as: Low consumption of fruit, vegetables, milk, seafood, and high consumption of red meats, processed beef, and sugar sweetened drinks [4].

The appearance and growth of fast food has been a key factor in the change of the food system and the increase of these diseases. Popkin and Reardon (2018) estimate that in Mexico, the growth in sales from fast food restaurants has been of 7% between 2011 and 2014, and chains like McDonald’s that currently have 500 stores in 87 cities, dominate the market [5]. If the current trend continues, there are no signs that point to a reduction of these products in the market, nor the diseases associated with their consumption. On the contrary, this situation could get worse in the upcoming years. In Mexico, it is estimated that less than 25% of the population, of all group ages and gender, follow the Mexican dietetic recommendations (excluding the adherence to red meats that is above 75%) [6] and more than 64% consume more sugar than recommended. Further, more than 78% of children (ages 1–11 years) and more that 66% of adolescents (12–19 years) consume more than the recommended intake of saturated fats [7]. Moreover, the intake of discretional products (products with high content of saturated fats and/or added sugar or sugary drinks) is 12% to 26% higher in street food compared to food cooked at home. One study with a sample of kids and teenagers between 5–15 years old in the state of Morelos, showed that 87% of the participants had a dietetic pattern with high content in fats and sugar (36% and 51% respectively) [8]. On the other hand, very little is known about the level and type of food intake outside home, like the fast food restaurant chains, movie theaters, or street stands, mostly because the information collected in national health surveys have not made clear conclusions regarding this type of data.

In the United States, there are estimates that between 2007 and 2008, 33% and 41% of children and adolescents respectively, consumed food products and non-alcoholic drinks in fast food restaurants. It also showed that the child population had an average intake of 25% of their daily calories in these types of restaurants and other similar local businesses [9,10]. The consumption trends show increases, moving from 3% to 18% of the energy intake of these food products between 1977 and 2006 [11]. Usually the food products sold at these type of fast food restaurants do not meet the dietetic recommendations and they are not required to offer nutritional information in an effective way so that consumers can choose between healthier options [12]. These types of products also contain what is known as empty calories, which come from solid fats and added sugar, and usually exceed the recommendations established for this age group. According to the American Dietetic Guide, it is recommended that empty calories do not exceed 8%–19% of total energy intake, depending on the total calorie needs of each individual [13,14].

An increased tendency to eat out of home, along with concerns about the adverse nutritional profile of foods and drinks in restaurants, has motivated the introduction of caloric labels. However, very few studies have been made about the caloric content and the level of consumption of food products offered by fast food restaurant chains and movie theaters. Even more, there are currently no mandatory regulations to inform consumers about the nutritional contribution of these foods to the diets of school age children. The studies that have been conducted, especially in the United States, often use information provided by the fast food restaurants through their menus or webpages [15,16,17,18] and very few of them do an independent bromatological analysis of the products and drinks. 

The purpose of the present analysis is to identify the nutritional quality of kids’ meals in fast food restaurants and their contribution to the diet of school age children. For the purposes of this study, we will refer to school-age children as those ranging from 3 to 14 years of age and more globally referred to as children and teenagers in general. 

## 2. Materials and Methods

A bromatological analysis was carried out for the kids’ menu or meals offered for kids, including two breakfasts, four meals, and two popcorn sizes in six fast food restaurants and two movie theaters between December 2015 and February 2016. These businesses were located in the downtown and westside areas of the Mexican city of San Luis Potosi, in the state of San Luis Potosi. The locations were selected using information provided by the National Institute of Geography and Statistics (INEGI), considering a perimeter of 2.485 miles from the Laboratory of Food Science and Biochemistry at the Autonomous University of San Luis Potosí (UASLP), where the analysis was done. The products selected from the kids’ menu from those available at that time were: Two breakfast meals and one happy meal from McDonald’s, one Burger King meal, two meals from Carl’s Jr., one combo from the movie theatre Cinepolis and another combo from Cinemex.

In the bromatological analysis, the weight of the solids were recorded with a Pioneer Ohaus food scale with a sensitivity of 0.001 grams; these were crushed, processed, and analyzed. In the liquids a sample was taken, it was adjusted to 1% and brix degrees were taken in all of them. Humidity analysis was considered, along with carbohydrate content, proteins and lipids through the Fehling—Soxhlet method (to determine sugar, including liquids), the Microkjeldahl method (proteins), the weight loss method, the Soxhlet method (fats), and Gerber (for ice cream fats).

We assumed the breakfast, lunch, and popcorn meals were consumed by school-age children during breakfast, lunch, and snack hours. In order to compare them with the daily requirements in kilocalories (kcal), we used the Nutritional Intake Recommendation for the Mexican population from Borges et al. (2008) using a mean of moderate physical activity for different groups of school age: Preschool (3–5 years old), school age (6–11 years old), and teenagers (12–14 years old). In order to compare the energy contribution by macronutrient, the total energy requirement per day was used considering a breakdown of 55% carbohydrates, 15% proteins, and 30% lipids [19].

The study result of the contents was compared with the mean nutritional recommendations for each mealtime and age group, considering: (a) Total energy requirements, and (b) by macronutrients (proteins, carbohydrates, and lipids). To test differences of means, the non-parametric Wilcoxon method was used in the statistical software program Stata version 14.

## 3. Results

### 3.1. General Characteristics of The Kids’ Meals 

The characteristics of the kids’ meals are shown in Table 1. In general, they are made of one main food product, one drink, and one complement that can either be a snack, candy, or a gift. The analyzed meals have as a main product one hamburger with fries and the snack buttered popcorn. The offered drink comes in a standard presentation of 355 milliliters (mL) and 330 mL only in the snack presentation; however, the drink can be refilled for lunch. In four of the meals and one of the snack combos, we found gifts. In the meals, we generally found toys and in snack combos, a collectible cup with colored pens. 

### 3.2. Nutritional Content in Kids’ Meals and Nutritional Contribution to The Diet of School-Age Children 

Table 2 shows the bromatological results of the analyzed meals for the content of carbohydrates, proteins, lipids, and total energy. The means for each mealtime are compared with the mean recommendations by age group and gender in Table 3. The results of the difference in means test and their statistical significance are also reported. The estimated means are statistically different from the mean recommendations for total energy and for these three macronutrients in the three mealtimes, except for lunch in middle-school children. 

Figure 1 shows the mean energy contribution of each menu item per mealtime in bromatological results. The main dish is the one that contributes the most, however, it is notable that at breakfast the drink can represent 38% of energy contribution, and at lunch, the side dish 28% of meal content. 

Figure 2 shows the contribution of each menu item to the recommended intake by mealtime and by school age group. It is notable that in case of snack, the main dish (popcorn), provides over 100% of the mean energy intake recommendations for the three school age groups. In particular, for preschool age group, this contribution could amount to 270%. At breakfast, drinks contribute significantly to the intake of all ages, between 43% and 21%. In the case of lunch, the side dishes are the second most important contribution to energy intake after the main course, between 29% and 58%.

Figure 3 shows that for all school age groups the caloric and almost all macronutrients contribution of almost all mealtimes is not only high but even above 100% or 200% of the mean recommended daily intake (reaching to more than 400% of the recommendations of carbohydrates and lipids of preschool age group), being higher for the snack meals and second highest for lunch meals (see Table 3 for more data information). On the contrary, in the case of proteins at breakfast (a), the mean contribution is below the recommended intake for all groups, at lunch (b) and snack (c) only for middle school age the same thing happened. 

## 4. Discussion

According to the results, the energetic contribution of meals intended for school-age children are above the nutritional recommendations for these meals. The estimations are consistent with some research studies done in the United States [18,20,21] and in Monterrey and Mexico cities [22].

Nutritional knowledge and understandable information are essential to change dietary behaviors [19,23]. However, not every restaurant provides nutritional information about the products they sell and doing so would help consumers make better decisions.

More extensive studies and research is required in this field to identify possible effective strategies that can allow consumers to select and eat food items with less calories, to thereby reduce the consumption of energy from fast food restaurants as one of the measures to improve health problems related to diet in Mexico [24,25]. Some recommendations include: 1. Reformulate the meals so that the items meet certain standards related to calories, fats and sugar; 2. Eliminate sodas and other sugar added drinks in kids meals; 3. Offer more fruit options (not just fruit juice) and vegetables and make them predetermined side dish in each kids meal; 4. Provide nutritional information for every single item on the menu; 5. Market and sell only healthy options for kids and teenagers through the many marketing strategies managed by the restaurant, including the media, webpages, store promotions, school related activities, and other places [26]. Faced with this situation, it is unlikely that *voluntary* efforts from these fast food restaurants to reduce portions sizes will be effective.

The use of integral approaches to reduce the energy intake coming from fast food is needed. One of the more documented strategies is clear and easy to understand labelling for food items and menus in these restaurants. This would require greater scientific evidence to evaluate the impact the regulation has on total calorie intake per day and the effects that it may have on this type of industry. Nutritional and public health policies could also be focused not only on encouraging the population to count their daily calories, but also importantly, to learn about the meal ingredients and menus, food processing, dietetic sources, and preparation methods. One challenge would be to determine an optimal approach to provide this information especially with those segments of our population where education and financial resources are more limited [27,28,29].

Some of the limitations found while conducting this analysis was that sugary drinks have free refill for lunch time and considering that the customer can get as much as wanted, the calculations made in this study for this mealtime may be underestimated for this type of drink. Even though this analysis takes a limited selection of products in one city, it should be considered that the restaurants selected for this research belong to restaurant chains that offer the same products in the rest of the country.

To our knowledge, this is the first study carried out in the city of San Luis Potosí. The findings presented here show that unhealthy food environments become normalized, especially for vulnerable groups like school-age children. This study demonstrates the extent to which this population exceeds the caloric and micronutrient recommendation per meal in fast food restaurants, as well as the potential that food environment regulations have to reduce the prevalence of overweight and obesity. Our results are of national significance because they contribute to an emerging evidence base that points to the need for mandatory food labeling at the point of sale to alert the consumer to what their kids are eating and promote the reformulation of these food product options. Therefore, this study contributes to information regarding the nutritional quality of fast food offered in Mexico and supports the need for more research in this area to be able to generate adequate public health proposals to improve diet quality among the Mexican population, specifically school-age children.

## 5. Conclusions

This research shows that the energetic content of breakfast, lunch and snack meals offered in fast food restaurants and movie theaters are above the recommended daily intake for school-age children. Regulations are urgently needed so that places that sell food and drinks offer nutritional information about their products, in a way that allows consumers and in particular, parents, to make informed decisions that align with the recommended intake for their kids. This would surely help to improve diet quality and may also have an impact on the prevalence of overweight and obesity in this vulnerable population.

## Figures and Tables

**Figure 1 nutrients-12-00612-f001:**
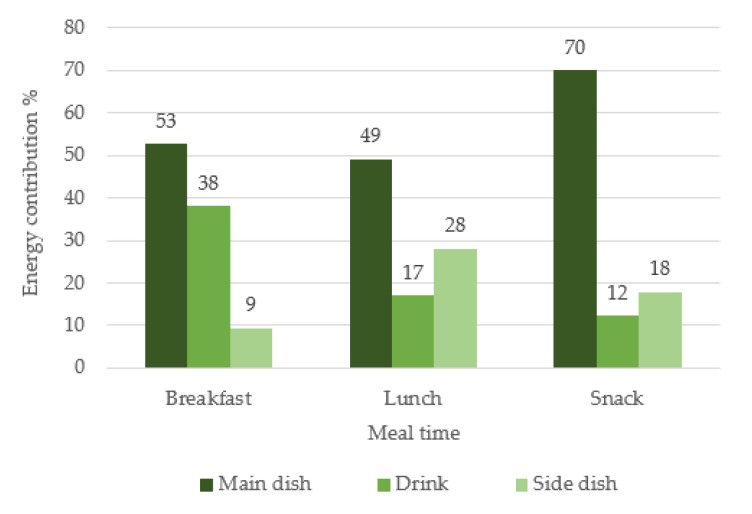
Percentage energy contribution of each menu item per mealtime (%).

**Figure 2 nutrients-12-00612-f002:**
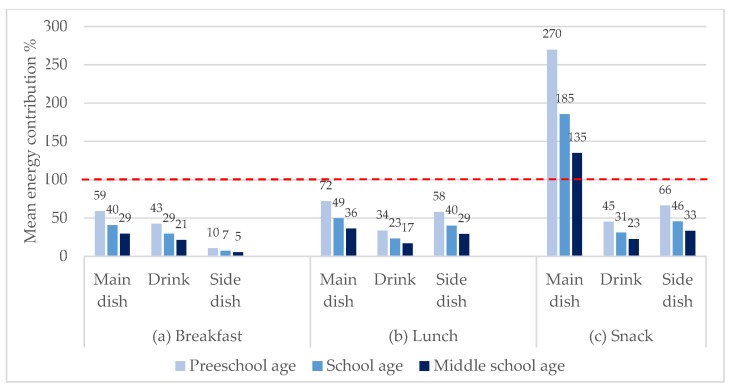
Percentage energy contribution of each menu item by school age per mealtime (%).

**Figure 3 nutrients-12-00612-f003:**
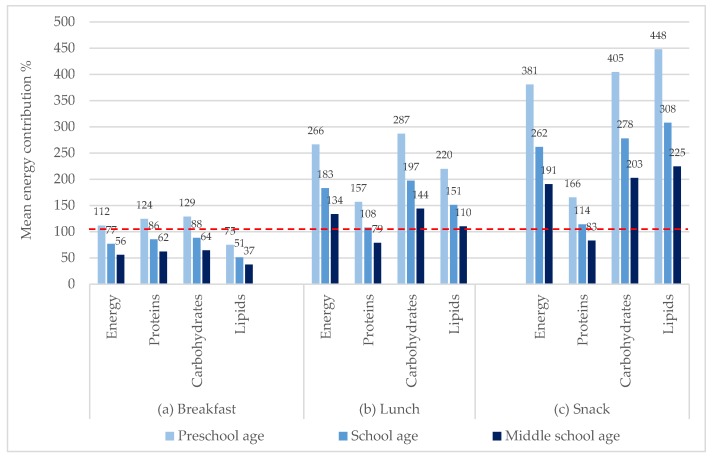
Mean energy and macronutrient contribution of kid’s meals regarding the mean recommended requirements for (**a**) breakfast, (**b**) lunch, and (**c**) snack, by school age (%).

**Table 1 nutrients-12-00612-t001:** Characteristics of kids’ meals sold at fast food restaurants and movie theaters in the city of San Luis Potosí.

Kids Meals	Breakfast		Lunch				Snack	
	Breakfast 1 McDonald´s	Breakfast 2 McDonald´s	Lunch 1 Burger King	Lunch 2 Carl’s Jr. with Toy	Lunch 3 Carl’s Jr. with Ice Cream	Lunch 4 McDonald´s Happy Meal	Kid meal 1 Cinépolis	Kid meal 2 Cinemex
**Kids’ meals characteristics**								
Menu items/combo	Pancakes (3pcs), juice or coffee, strawberry yoghurt	Ham and eggs (2 pcs), juice or coffee, strawberry yoghurt	Hamburger, fries, and soda	Hamburger, fries, and soda	Hamburger, fries, soda, and ice cream	Hamburger, fries, and soda	Buttered popcorn, soda, and gummies	Butter popcorn, flavored drink, marshmallow with chocolate
Main item content (g)	121	110	109	168	168	116	91	163
Drink content (mL)	355	355	355	355	355	355	355	330
Energy content drink (kcal)	136	136	146	114	114	142	142	30
Total energy content (kcal)	359	355	612	831	1,923	709	692	764
**Side item characteristics**								
Soda with refill (Yes/No)	No	No	Yes	Yes	Yes	Yes	No	No
Candy (Yes/No)	No	No	No	No	No	No	Yes	Yes
Meal gift (Yes/No)	No	No	Toy	Toy	Ice cream	Toy	No	Collectible cup with color pencils
**Sale characteristics**								
Hours of sale	9:00–12:00	9:00–12:00	12:00–close	11:00–close	11:00–close	13:00–close	11:00–22:30	11:00–22:30
Kids meal (Yes/No)	Yes	Yes	Yes	Yes	Yes	Yes	Yes	Yes
Playground (Yes/No)	Yes	Yes	Yes	Yes	Yes	Yes	No	No
Party area (Yes/No)	No	No	Yes	Yes	Yes	Yes	No	No

Source: Information obtained during the field work and laboratory time.

**Table 2 nutrients-12-00612-t002:** Bromatological results for energy and macronutrient content of kids’ meals.

Kids´Meals	Nutritional Content (kcal)			
	Carbohydrates ^1^	Proteins ^1^	Lipids ^1^	Total Energy
Breakfast				
Breakfast 1 McDonald´s	291	54	13	359
Breakfast 2 McDonald´s	160	65	130	355
Lunch				
Lunch 1 Burger King	337	83	191	612
Lunch 2 Carl’s Jr. with toy	432	114	286	831
Lunch 3 Carl’s Jr. with ice cream	1232	93	308	1923
Lunch 4 McDonald´s Happy Meal	414	70	225	709
Snack				
Kid meal 1 Cinépolis	476	40	180	692
Kid meal 2 Cinemex	375	55	334	764

Source: Mean content estimated from the analysis. (1) Mean energy contribution considering a breakdown of 55% carbohydrates, 15% proteins, and 30% lipids according to the total daily energetic requirements by school group age and gender.

**Table 3 nutrients-12-00612-t003:** Comparison between nutritional content of school-age children’s meals with daily intake recommended by age groups (Kcal).

Mealtime and School Age	Energy		Carbohydrates				Proteins			Lipids		
	Study Result (kcal) ^1^	RDI (kcal) ^2,3,4^	Study Result (kcal) ^1^		RDI (kcal) ^2,3,4^		Study Result (kcal) ^1^	RDI (kcal) ^2,3,4^		Study Result (kcal) ^1^	RDI (kcal) ^2,3,4^	
Breakfast												
Preschool age	357	319 (255,383)	*	226	175 (140,210)		60	48 (3857)	***	72	96 (77,115)	
School age		464 (371,556)	*		255 (204,306)			70 (5683)	**		139 (111,167)	**
Middle school age		636 (509,763)	*		350 (280,420)	**		95 (76,115)	***		191 (153,229)	*
Lunch												
Preschool age	1019	383 (319,446)	*	604	57 (4867)	***	90	210 (175,245)	***	253	115 (96,134)	***
School age		556 (464,649)	*		83 (7097)			306 (255,357)	**		167 (139,195)	***
Middle school age		763 (636,890)			115 (95,134)	***		420 (350,490)			229 (191,267)	
Snack												
Preschool age	728	191 (126,255)	*	426	29 (1938)	***	48	105 (70,140)	***	257	57 (3877)	***
School age		278 (185,371)	*		42 (2856)			153 (102,204)	***		83 (56,111)	***
Middle school age		382 (254,509)	*		57 (3876)	*		210 (140,280)	***		115 (76,153)	***

Source: Mean content estimated from the bromatological analysis and Bourges et al. (2008). (1) Mean estimation from Table 1. (2) RDI: Recommended daily intake in kilocalories (kcal). (3) Moderate activity averages estimated from recommendations in Appendix A (+/− 5%). (4) Difference test of means at 95% confidence and levels of significance: *** *p* < 0.01, ** *p* < 0.05, * *p* < 0.10.

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
