# Peer review of "Nutritional Quality of Fast Food Kids Meals and Their Contribution to the Diets of School-Aged Children"

_nutrients, 2020, doi:10.3390/nu12030612_

Round 1

Reviewer 1 Report

This study undertakes an analysis of the biochemical composition of popular fast foods targeted at young children and teens in Mexico. I think this sort of analysis is certainly valuable as it provides a more accurate breakdown of true nutrient composition of the foods, instead of relying on nutritional tables provided by the fast food chain. However, the authors may have missed an opportunity in this study to compare biochemically determined nutrient composition versus marketed nutrient composition. This would be of great interest to readers. I would suggest the researchers seek assistance with language editing as this paper was incredibly difficult to read and I found myself often having to read sentences twice or three times just to understand what message was being conveyed. Much of the introduction can also be reduced, as there is no real need to go into too much detail about the scale of the childhood obesity epidemic and its health consequences (this is, in a journal like nutrients, assumed knowledge for many readers). I would also suggest the authors reconsider how they might present the comparative data between energy/macronutrient composition of the meals versus the calculated nutrient recommendations/requirements of each age and gender group. At the moment, it is quite difficult to read across such a large table and perhaps a bar chart showing how much (%) each meal exceeds the calulcated nutrient requirements is easier to interpret.

Author Response

Nutrients

Dear Editor-in-chief

In order to process this manuscript further, we expose point by point our answers and the details of the revisions in the manuscript in accordance with the reviewers’ comments.

Request/Comment

Response / Change

Localization

Reviewer 1

The authors may have missed an opportunity in this study to compare biochemically determined nutrient composition versus marketed nutrient composition. This would be of great interest to readers.

The marketed nutrient composition was not available at the points of sale where the product samples were taken. It is precisely a vacuum in the regulation, which is not required to express it and is the suggestion of public policy that emerges from the article.

All the document

I would suggest the researchers seek assistance with language editing as this paper was incredibly difficult to read and I found myself often having to read sentences twice or three times just to understand what message was being conveyed.

We appreciate this comment. The manuscript was reviewed for a second native English speaker. We hope we have improved this aspect of the document.

All the document

Much of the introduction can also be reduced, as there is no real need to go into too much detail about the scale of the childhood obesity epidemic and its health consequences (this is, in a journal like nutrients, assumed knowledge for many readers).

We reduced the introduction, especially we removed the information about the children obesity epidemic and its health consequences.

Introduction section

I would also suggest the authors reconsider how they might present the comparative data between energy/macronutrient composition of the meals versus the calculated nutrient recommendations/requirements of each age and gender group.

We restructure Tables 2 and 3 and create Tables 2, 3 and Supplementary Table S1. We hope to have improved the presentation of the results in this way.

Table 1, 2 and Supplementary Table S1

At the moment, it is quite difficult to read across such a large table and perhaps a bar chart showing how much (%) each meal exceeds the calulcated nutrient requirements is easier to interpret.

We changed the bar chart to make it easier to interpret.

Figure 1, 2 and 3

Reviewer 2 Report

Manuscript presented by Lucio et al. concerns a very important and at the same time complex topic of nutrimental quality ready-to-eat food offered in fast-food restaurants and their contribution to the diet of children.
Thank you for allowing me to review your study.
While the concept is interesting and the idea worth pursuing, as it stands it is very difficult to read and understand what it is that you actually did.
You talk about infants (in the title, introduction, aim) but you rather mean children and adolescents (look at the tables and figures)???
The presentation of results in tables and figures needs to be improved.
What are you actually comparing? Why is p-value equal to 0?
Figures are a repetition of the data.
You write about chemical analysis (in materials and methods ) and then in Table 3 you write "Estimated mean content b (Kcal)"? Explain what you really did?
I propose to expand the results with more data - a larger range of meals, kids' packs and nutrients, e.g. sugar, sodium and the addition of vegetables and fruit to meals. This will allow for a better justification for the creation of recommendations for population and restaurant.
Indicate what is the strength of your study.
Moreover, this manuscript is not prepared in accordance with the submission guidelines. You have to do according to MPDI guidelines.
So I have to say I cannot adequately assess this paper as it is currently written. You really have to be clearer about what you did.
I suggest that you have the manuscript edited for English usage as well.

Author Response

Nutrients

Dear Editor-in-chief

In order to process this manuscript further, we expose point by point our answers and the details of the revisions in the manuscript in accordance with the reviewers’ comments.

Reviewer 2

While the concept is interesting and the idea worth pursuing, as it stands it is very difficult to read and understand what it is that you actually did. 

We hope the changes to the document, tables and figures responds your comment.

All the document, Tables and Figures

You talk about infants (in the title, introduction, aim) but you rather mean children and adolescents (look at the tables and figures)

We introduce the term "school-age children" in title, introduction, objective, tables and figures to refer children and adolescent between 3-14 age. We hope it can be clearer what are the group of age we analyzed.

All the document, Tables and Figures

The presentation of results in tables and figures needs to be improved.

We change tables and figures.

All tables and figures

What are you actually comparing? Why is p-value equal to 0? 

We are comparing the results of our study estimates with the recommendations by school-age group, gender and meal time (now Table 3).

It means that the value is very low, with many "0" before any other digit.

So the interpretation would be that the results are significant. Therefore p=0.0000 implies high significance.

Table 3

Figures are a repetition of the data.

We change all the figures hoping this new version of the results being more clear and no repetitive.

Figures 1, 2 and 3

You write about chemical analysis (in materials and methods) and then in Table 3 you write "Estimated mean content b (Kcal)"? Explain what you really did?

In the fourth paragraph of Material and Methods and in the new table 3 it is said "Study result" which shows the estimates of the study of the energy and macronutrient content of the meals. We hope this clarifies what was done.

Fourth paragraph of Material and Methods and Table 3

I propose to expand the results with more data - a larger range of meals, kids' packs and nutrients, e.g. sugar, sodium and the addition of vegetables and fruit to meals. This will allow for a better justification for the creation of recommendations for population and restaurant.

We would like to have time and resources to expand the sample, however, at this time, it is not possible to do so. However, we suggest in the discussion section that more extensive studies be made of the contents of kids' meals in fast-food restaurants.

Fourth paragraph of Discussion

Indicate what is the strength of your study.

We add the strength to our study

Last paragraph of Discussion

Moreover, this manuscript is not prepared in accordance with the submission guidelines. You have to do according to MPDI guidelines.

We make sure that citations within the text are in the correct format; references 
at the end of the text are in the correct format; and figures and/or tables are 
placed at appropriate positions within the text and are of suitable quality.

Tables are prepared in MS Word table format, not as images.

Introduction and Discussion

All references page 13-15.

You really have to be clearer about what you did. 

We hope the changes to the document, tables and figures responds your comment.

All document

I suggest that you have the manuscript edited for English usage as well.

We appreciate this comment. The manuscript was reviewed for a second native English speaker. We hope we have improved this aspect of the document.

All document

Round 2

Reviewer 2 Report

Although the authors have put a lot of work into the correction of this manuscript, it still requires further improvement.
Particular attention should be paid to tables and figures.
There are still a lot of inaccuracies to improve, e.g. in table 2 and 3 Nutrimental content (kcal) - Carbohydrates (kcal), Proteins (kcal), Lipids (kcal) - ???
Table 3 instead of population groups - rather gender; Carbohydrates kcal - rather g.; Furthermore table headers, explanations, p-vale, y-axis labelling missing- figure 1-3.
The vocabulary should be harmonized in the whole work,
I also propose to add more data to the abstract.

All work still requires good reorganization and proofreading.

Author Response

Cover letter

Nutrients

Dear Editor-in-chief

In order to process this manuscript further, we expose point by point our answers and the details of the revisions in the manuscript in accordance with the reviewers’ comments of Round 2. We have cited the page, line number, section and exact change, so that the editor can check the changes expeditiously, as you ask to us.

Page

Line

Section

Change

Round 2: Review 2

Reviewer’s comment a) Particular attention should be paid to tables and figures

General response of authors: Improvements were made in tables and figures. You can also see the answer of the Reviewer’s comment b), below.

4

119

Results

Table 1. Main content (gr) was replaced by Main content (g); We added “(Yes/No)” after soda with refil, candy, meal gift, kids’meal, playground, and party area.  

Reviewer’s comment b) There are still a lot of inaccuracies to improve e.g. in table 2 and 3 Nutrimental content (kcal)- Carbohydrates (kcal), Proteins (kcal), Lipids (kcal)-???. Table 3 instead of population groups – rather gender; Carbohydrates kcal – rather g.; Furthermore table headers, explanations, p-value, y axis labelling missing – figure 1-3.

General response of authors: Improvements were made in Table 2, Table 3, Figure 1, Figure 2 and Figure 3.

4

119

Results

Table 1. We replaced “w/toy” and “w/ice cream” by “with toy” and “with ice cream”.

4

119

Results

Table 1. We replaced “main item” to “main dish” to standarize the manuscript.

5

128

Results

Table 2. The title “Energy and macronutrient content of kids' meals” was replaced by “Bromatological results for energy and macronutrient content of kids' meals”. We added the word “with” to the lunch 2 and 3 to be more specific and clear. The title is in the center of the table.

5

128

Results

Table 2. We added the word “with” to the lunch 2 and 3 to be more specific and clear. The title is in the center of the table.

6

131

Results

Table 3. We improved the format. We added “kcal” after RDI, and Study result to be clear about the units. We have included the note: RDI: Recommended Daily Intake in kilocalories (kcal) to give coherency and improve the interpretation.

6

131

Results

Table 3. About the results by gender, we only present the mean results for both, girls and boys.

7

141

Results

Figure 1. Was improved. Title was replaced by “Percentage energy contribution of each menu item per meal time (%)”.

7

141

Results

Figure 1. We replaced the word “main item” to “main dish”.

7

151

Results

Figure 2. The title was replaced by “Percentage energy contribution of each menu item by school age per meal time (%)”.

7

151

Results

Figure 2. The organization of the information was changed since Round 1; we improved also the presentation of the graphic. We have gathered the three figures in a single presentation.

8

161

Results

Figure 3. The title was replaced by “Mean energy and macronutrient contribution of kid's meals regarding the mean recommended requirements for (a) Breakfast, (b) Lunch, and (c) Snack, by school age (%)”.

8

161

Results

Figure 3. We improved the presentation of the graphic. We have gathered the three figures in a single presentation.

6,7 y 8

141, 151, 161

Results

Figure 1, 2 y 3. Axis labelling was included.

Reviewer’s comment c): The vocabulary should be harmonized in the whole work, I also propose to add more data to the abstract. All work still requires good reorganization and proofreading.

General response of authors: Changes were made. We have checked the vocabulary of all the work to have better cohesion, and harmonization. We also have changed some words to be more clear, giving coherency and improve the cohesion of the text. We hope we have improved this aspect of all the document and to have a better structure and presentation of the results.

1

16

Abstract

Was improved with language editing.

We changed the words “infant and teenager” by “children and adolescents”.

1

22

Abstract

We have added more information to improve the message.

1

27

Abstract

We changed the sentence “about the nutrimental standards should exist for the commercialization of these type of food products for infant and teenager population” by “regarding nutritional standards should exists for these types of commercialized food products for school-age children and adolescents”.

1

30-31

Keywords

“Caloric input” was replaced by “caloric contribution”; and we added children, and a “.” after the word obesity.

2

49

Introduction

“Mexican dietetic suggestions” was changed by “Mexican dietetic recommendations”.

2

50

Introduction

We added the word “Further” and putted a point before to give more cohesion to the text. We eliminated “kids” and changed by ”children” to specify the age; also changed the word teenagers by adolescents and we added the age. 

2

59

Introduction

“Infant” was replaced by “children”; and “teenager population” by “adolescents”

2

61

Introduction

“Infant” was replaced by “child”

2

83

Materials and methods

“A bromatological analysis was made in the menu or meals for kids in two breakfasts” was replaced by “A bromatological analysis was carried out for the kids menu or meals offered for kids, including two breakfasts”.

3

93

Materials and methods

“The solids was made” was replaced by “the solids were recorded”

3

99

Materials and methods

“Infant population” was replaced by “school-age children”.

3

101

Materials and methods

“it was used the Nutrimental Intake Recommendation for the Mexican Population using an mean” was replaced by “we used the Nutritional Intake Recommendation for the Mexican population using a mean.

3

106

Materials and methods

“The contents analyzed were compared was replaced” by “The study result of the contents”. And “nutrimental” was replaced by “nutritional”. Study result is the term that is used in tables.

4

118

Results

Table 1. We replaced “kids pack” by “kids meal” to the harmonization of the vocabulary.

5-6

127, 130

Results

The word “average” was replaced by “mean” in tables 2 y 3.

5-6

128, 131

Results

Table 2 and Table 3. The word “snacks” was replaced by “snack”

10

218

Supplementary material

“Nutrient content” was replaced by “nutrimental content”.

Words were changed in singular: “snack”.

Authors’ revision

11

228

Author contributions

We eliminated our own description because it was repetitive with the one of the template of the submission.

Kind Regards, 

Dr. Lizbeth Tolentino-Mayo, PhD

[email protected]

+52 (555)487-1027
